Enhancing georeferenced biodiversity inventories: automated information extraction from literature records reveal the gaps

Kopperud Bjørn Tore 1 2 3
http://orcid.org/0000-0002-0446-4705 Lidgard Scott 4
http://orcid.org/0000-0002-3732-6069 Liow Lee Hsiang 1 5 l.h.liow@ibv.uio.no
1 Natural History Museum, University of Oslo , Oslo , Norway
2 GeoBio-Center, Ludwig-Maximilians-Universität München , München , Germany
3 Department of Earth and Environmental Sciences, Ludwig-Maximilians-Universität München , München , Germany
4 Negaunee Integrative Research Center, Field Museum of Natural History , Chicago, Illinois , U.S.A.
5 Centre for Ecological and Evolutionary Synthesis, University of Oslo , Oslo , Norway
Edwards Scott
Electronic publication date: 2022 Aug 18
Publication date: 2022
Volume: 10
Electronic Location ID: e13921
Received 2022 Apr 22; Accepted 2022 Jul 29
Copyright: © 2022 Kopperud et al.
Copyright year: 2022
Copyright holder: Kopperud et al.
License: This is an open access article distributed under the terms of the Creative Commons Attribution License, which permits unrestricted use, distribution, reproduction and adaptation in any medium and for any purpose provided that it is properly attributed. For attribution, the original author(s), title, publication source (PeerJ) and either DOI or URL of the article must be cited.
License URL: https://creativecommons.org/licenses/by/4.0/

Keywords: Marine invertebrates, Bryozoa, Geographic distribution, Latitudinal diversity gradient (LDG), Public data repositories, Natural langauge processing (NLP), Text-mining, Bimodality, Species richness

Funding: European Research Council (ERC) European Union’s Horizon 2020 research and innovation programme 724324 This project is supported by the European Research Council (ERC) under the European Union’s Horizon 2020 research and innovation programme (grant agreement No. 724324 to Lee Hsiang Liow). The funders had no role in study design, data collection and analysis, decision to publish, or preparation of the manuscript.

==============================
We use natural language processing (NLP) to retrieve location data for cheilostome bryozoan species (text-mined occurrences (TMO)) in an automated procedure. We compare these results with data combined from two major public databases (DB): the Ocean Biodiversity Information System (OBIS), and the Global Biodiversity Information Facility (GBIF). Using DB and TMO data separately and in combination, we present latitudinal species richness curves using standard estimators (Chao2 and the Jackknife) and range-through approaches. Our combined DB and TMO species richness curves quantitatively document a bimodal global latitudinal diversity gradient for extant cheilostomes for the first time, with peaks in the temperate zones. A total of 79% of the georeferenced species we retrieved from TMO (N = 1,408) and DB (N = 4,549) are non-overlapping. Despite clear indications that global location data compiled for cheilostomes should be improved with concerted effort, our study supports the view that many marine latitudinal species richness patterns deviate from the canonical latitudinal diversity gradient (LDG). Moreover, combining online biodiversity databases with automated information retrieval from the published literature is a promising avenue for expanding taxon-location datasets.

Introduction

Biogeography

Global biogeographical and macroecological studies require data on aggregate entities, such as location-specific occurrences of taxa and regional species assemblages, in order to understand emergent patterns at global and/or temporal scales (McGill, 2019). Assembly of such detailed yet broad-scale data is highly labor-intensive; the sampling effort required for a specific research question can be daunting for any one researcher or single research team. This is one reason why collaborative and often public databases have gained traction (Klein et al., 2019; Heberling et al., 2021). For instance, empirical global biogeographic analyses (Costello et al., 2017; Rivadeneira & Poore, 2020; Chaudhary et al., 2021; Hughes et al., 2021) are increasingly based on public databases of georeferenced taxonomic occurrences, such as the Ocean Biodiversity Information System (OBIS, www.obis.org) and the Global Biodiversity Information Facility (GBIF, www.gbif.org). Analyzing such georeferenced databases with tools that partially alleviate incomplete or biased sampling (Edgar et al., 2017; Kusumoto et al., 2020; Zizka et al., 2020; Grenié et al., 2022), allows us to address questions on large-scale distributions of clades, especially those that are well-represented in such databases. For less well-studied clades, however, prospects for obtaining large amounts of such data are lower. Answering pattern-based questions such as ‘how many species of clade z are found in location y’ and more process-oriented questions such as ‘how did the current latitudinal diversity gradient form’ both require location-specific taxonomic data in substantial volume. In addition, generalized biogeographic hypotheses have the potential to be supported more robustly if they include a greater diversity of clades.

Automated information retrieval

Automated information retrieval (Hirschberg & Manning, 2015) is one recent approach to complement the time-consuming manual activity of data compilation from the scientific literature. We seek to advocate for the enormous interdisciplinary potential of automated information retrieval, while acknowledging that it is a rapidly developing area at an early stage of development whose capabilities are just beginning to be realized. Automated text-mining is well-established in the biomedical realm (Percha, Garten & Altman, 2012; Christopoulou et al., 2020), but has only recently been adopted for biodiversity studies (Peters, Husson & Wilcots, 2017; Kopperud, Lidgard & Liow, 2019). As far as we are aware, automated text-mining has never been applied to the literature for extraction of taxon occurrences in given locations for the purpose of understanding biogeography (but see Page, 2019). Natural language processing is primarily concerned with understanding human language. This can involve a range of simple tasks, for example sentence delimitation and tagging part-of-speech (identifying nouns, verbs, adjectives etc.), to more complex tasks such as language translation, text generation, or text-to-speech. We use natural language processing tools (De Marneffe et al., 2014; Bojanowski et al., 2017) to facilitate a task known as relation extraction (Wang et al., 2022), which involves understanding relations among words in a sentence. We use these extracted relations to compile cheilostome text-mined occurrence data (TMO), which we compare with data from DB.

Cheilostomes

Cheilostome bryozoans, though less well-studied than several metazoan clades of similar size, are ubiquitous in benthic marine habitats. They are the most diverse order of Bryozoa with a conservatively estimated 4,921 extant described species (Bock & Gordon, 2013), or 83% of all Bryozoa. Bryozoans are ecologically important habitat builders (Wood et al., 2013) and are vital components of the marine food chain (Lidgard, 2008). Despite important analyses of regional species distributions (Clarke & Lidgard, 2000; López Gappa, 2000; Barnes & Griffiths, 2008; Hirose, 2017, Boonzaaier-Davids, Florence & Gibbons, 2020; Denisenko, 2020), their global species richness distribution has never been quantified. We argue that even with concerns about the incompleteness of public records for the purpose of inferring regional to global diversity patterns (e.g., Klein et al., 2019; Chollett & Robertson, 2020, Moudrý & Devillers, 2020; Hughes et al., 2021), it is worth exploring cheilostome data in such public databases. We do so in order to identify spatial gaps in sampling but also to ask if automated information retrieval can enhance the species occurrence data available in public databases, specifically OBIS and GBIF (henceforth shortened as DB).

Taxon occurrence

Taxon occurrence data from DB and TMO are not expected to be the same. DB records are comprised of diverse sources, but the bulk of contributions come from research institutes that are willing to digitize and organize field notes and collections (Saran et al., 2022). The bulk of TMO data are, on the other hand, based on published taxonomic literature. We ask if they could, separately or in combination, shed light on a long-standing biogeographic hypothesis in the bryozoological literature. Many different groups of organisms show the canonical LDG, a species richness peak in tropical regions and decreasing species richness towards the temperate and polar zones (Hillebrand, 2004; Menegotto, Kurtz & Lana, 2019). Despite being common across marine and terrestrial realms, and among diverse eukaryote clades, the LDG is not universal (Chaudhary et al., 2021). Marine extratropical bimodal species richness peaks have been observed, for example in brittle stars (Woolley et al., 2016), polychaetes (Pamungkas, Glasby & Costello, 2021), crustaceans (Rivadeneira & Poore, 2020), fishes (Lin et al., 2021), and brown macroalgae (Fragkopoulou et al., 2022), among other groups. Bimodality has also been suggested for cheilostome bryozoans (Schopf, 1970; Clarke & Lidgard, 2000; Barnes & Griffiths, 2008).

The TMO and DB data in combination support the view that the latitudinal diversity pattern of living cheilostomes is bimodal. These data reveal highest levels of estimated species richness in temperate latitudes, but TMO species richness has a peak in the northern hemisphere while DB has a peak in the temperate south. Moreover, two datasets differ significantly in the geographic richness patterns in Atlantic vs. Pacific ocean basins (Schopf, 1970; Barnes & Griffiths, 2008). We discuss the pros and cons of TMO and public databases such as OBIS and GBIF and how their differences can help us understand the uncertainties of the retrieved spatial diversity patterns, beyond what is estimated within the confines of each dataset.

Methods

DB data retrieval

We used the R-package robis (Provoost & Bosch, 2021) to access OBIS (2022), and the web interface of GBIF to retrieve latitude/longitude occurrence records of cheilostomes (GBIF dataset; DOI 10.15468/dl.58pd9h). Both databases were accessed on 21.02.2022. We removed records without species epithets. For taxonomic ambiguities such as cf., aff., we disregard the uncertainty; for instance, Microporella cf. ciliata becomes Microporella ciliata. Records with genus names that are not accepted according to either the Working List of Genera and Subgenera for the Treatise on Invertebrate Paleontology (DP Gordon, 2019, personal communication), World Register of Marine Species (WoRMS Editorial Board, 2022) or www.bryozoa.net (Bock, 2022) were also removed. For all unaccepted species names that are found in WoRMS, we translated the species name to the accepted species name according to WoRMS. We also dropped all Linnean binomials that were not found in WoRMS. The result is 831 unique genus names and 4,549 unique genus-species combinations (henceforth simply species) in 149,042 retained OBIS and GBIF records.

TMO (Text-Mined Occurrence) data retrieval

We follow a previously detailed text-mining procedure (Kopperud, Lidgard & Liow, 2019) with modifications. We extracted text from two collections of published works, our own corpus (3,233 pdf documents) and the GeoDeepDive archive (GDD, https://geodeepdive.org/), which contains full-text contents of journal articles. Only English language publications and those likely to feature extant bryozoans were used for information extraction (see Appendix S1). The natural language processing steps are detailed in the following sections, but we summarize them here. First, we perform a linguistic annotation task on the sentences of the text. Second, we extract so-called “candidates”, that is, pairs of location names and species names that co-occur in the same sentence. Third, we develop a location-name verifier to remove several mis-identified location names. Finally, we use a relational classifier that tells us whether the sentence actually said that the species was in the location.

Linguistic annotation

We used CoreNLP (Manning et al., 2014) for an initial natural language analysis, including tokenization, named-entity recognition, and dependency grammar annotation (Hirschberg & Manning, 2015). Tokenization splits text into tokens, which are words and punctuation, after which sentences can be demarcated. In order to recognize bryozoan species names, we first assembled a list of known species names, based on the online compendia www.bryozoa.net and WoRMS. We used these names to assemble a set of rules (TokensRegex, Chang & Manning, 2014) that recognize relevant names of species. Finally, we applied CoreNLP to assign dependency grammar (i.e., relations among words in a sentence, De Marneffe et al., 2014), which we used as a feature for the relation classifier explained later. We used a generic, pre-trained machine-learning model (Finkel, Grenager & Manning, 2005) to recognize location names in the text. We found that this generic named-entity recognition was prone to detect false positives such as author names (e.g., “Hincks” or “Darwin”). In order to remedy this, we trained a machine-learning classifier to verify whether the tagged entities were indeed proper location names (see section “Location name verification”).

Candidate extraction

Consider the sentence below, as an illustration (from Tilbrook, Hayward & Gordon, 2001, p. 50):

“The avicularia resemble those seen in B. intermedia (Hincks, 1881b), from Tasmania and New Zealand, but this species is only just over half the size of B. cookae.”

This sentence contains two species names (“B. intermedia” and “B. cookae”) and two location names (“Tasmania” and “New Zealand”). Each species-location pair is a candidate relation. The sentence implies that B. intermedia is found in New Zealand (a positive relation), but does not say anything about where B. cookae is found (a negative relation).

Whenever we found an abbreviated genus name, such as “B.”, we searched for genus names in the current and 14 previous sentences. In reverse chronological order, we looked for any span (i.e., one or more consecutive tokens) tagged as being a taxon that starts with the same capital letter, and chose the first genus name for the de-abbreviation (here “Beania”).

Neural network

Two of our classifiers are implemented as neural networks, and so we give a brief introduction for readers who are not familiar with this topic. A neural network can be thought of as a computational graph, with inputs, outputs, and several operations in the middle. For organizational purposes, we refer to sections of these operations as “layers”. Each layer has a set of associated coefficients or weights that are either set a priori or initialized randomly. For a set of inputs and labels, it is possible to evaluate a loss function that estimates how much the output diverges from the assigned label. Next, one can employ a back-propagation technique (Rumelhart, Hinton & Williams, 1986) to compute partial derivatives or gradients for the loss value with respect to each coefficient in the network. These gradients, coupled with an algorithm for gradient descent, can be used to learn the coefficients, and consequently minimize the loss of the network during training. We split the annotation data in three parts: training-, validation- and test sets (roughly 80, 10, 10% candidates per set, respectively). We used the training data to learn the coefficients of the network. We used the validation data to fit the hyperparameters (e.g., learning rate, layer dimensions) and to decide when to stop training. By evaluating the trained network on the test data we get an estimate of how well the classifier performs on out-of-sample sentences. In order to remove possible confounding effects of similar sentences in a publication, we made sure that all candidates from a single publication were contained within only one of the training, validation or test datasets.

Location-name verification

The generic named-entity recognition was prone to detecting false positives. To reduce false positives here, we evaluated 6,000 candidates and annotated whether the assigned location name was indeed a location name or a false positive. We used these annotations to train a machine-learning classifier; a neural network implemented in Keras (Chollet et al., 2015). For the previously mentioned classifiers, we used reference implementations and did not change the default settings considerably.

Our location-name classifier received two inputs: the sequence of tokens in the candidate’s sentence, and a sequence of indicator variables denoting where the location name was in the sentence. The inputs enter into embedding layers: the tokens were embedded in a 300-dimensional vector space using a pre-trained fastText model (Bojanowski et al., 2017), and the indicator variables were embedded as two orthogonal unit vectors. The coefficients for the embedding layers were constrained to be static during training. After concatenation, the embedded values were fed to a Long Short-Term Memory (LSTM, Hochreiter & Schmidhuber, 1997) recurrent layer, with dropout values of 0.2. Since the sentences were of variable length, we padded the shorter sentences and used masking to avoid any processing of the padded features during training. We used the final time step of a second recurrent layer as input to a final two-dimensional hidden layer, with a softmax activation function. Since the softmax function maps all real inputs to outputs of [0,1], while ensuring that they sum to one, we interpreted the output as a probability mass. We used cross entropy as the loss function, and the ADAM algorithm (Kingma & Ba, 2014) for gradient descent. Unless otherwise stated, we used default settings in Keras for other parameters and options.

We used a test data set comprising 10% of the labelled candidates to evaluate several aspects of our machine-classifier: (i) accuracy, the ratio of correct predictions to all predictions; (ii) precision, the ratio of true positive predictions to all positive predictions; (iii) recall, the ratio of true positive predictions to all positive labels; (iv) false positive rate (FPR), the ratio of false positive predictions to all negative labels; and (v) F1, the harmonic mean of precision and recall. Each of these metrics yielded different information on the reliability of the extracted data.

For each epoch (an iteration in which the classifier sees the entire training set), we computed the F1 metric. We trained the classifier for 50 epochs, and saved the coefficients whenever the validation F1 was better than the previously best validation F1, effectively conditioning the learning on maximizing the F1 metric. The location-name classifier achieved an F1 of 94.7%, accuracy of 93.2%, recall of 97.4%, precision of 92.2%, and a false positive rate of 14.0% (see Fig. S6a) when evaluated on the test set.

Relation classifier

We manually annotated 4,938 unique candidates (species-location pairs) to form our training dataset. If the sentence explicitly stated or strongly implied that the taxon was found in the mentioned location, we labelled the candidate as ‘positive’. If not, we labelled the candidate as ‘negative’. These annotations were made by two persons, with intra- and inter-annotator accuracies of 91% (n = 200) and 85.6% (n = 211), respectively. This classifier is analogous to the location-name classifier (see previous section “Location-name verification”), except we used neither the indicator variables, nor the entire sentence. Instead, we used the sequence of tokens along shortest path in dependency grammar between the two spans in the sentence (see Xu et al., 2015; Kopperud, Lidgard & Liow, 2019). We trained our machine-classifier using the training set of these sequences to evaluate the classifier. This relation classifier achieved an F1 of 76.8%, accuracy of 73.1%, precision of 74.8%, recall of 78.9% and false positive rate of 34.3% (see Fig. S6b).

The estimated false positive rate of 34.3% is better than a random classifier baseline, but not as good as the false positive rates between human annotators at 14% and 16% (n = 200, assuming annotator A is correct, then evaluating annotator B, and vice versa). Similarly, the classifier accuracy of 73.1% is better than a weighted coin random classifier. If we use a random classifier that is as unbalanced as the labelled candidates (60% positives), this classifier gives an accuracy baseline of 52% (0.62 + 0.42 = 0.52). Yet, this is not as good as the intra- and inter-annotator accuracies at 91% and 85.6%, respectively. Incidentally, applying the relation classifier from Kopperud, Lidgard & Liow (2019) trained on relating taxon spans and geologic time interval spans, yielded virtually the same performance (evaluated on the same test data), despite not being trained on the specific task or having “seen” any location-name spans.

The relation classifier could potentially be improved by increasing the amount of training data, and/or using a more complex classifier, such as taking into account context-dependent word embeddings (Peters et al., 2018; Devlin et al., 2019). However, we believe that the major bottleneck is not lack of natural language understanding. Rather, the candidates themselves are not always linguistically sound, coherent and self-contained sentences. Specifically, much of the information that relays the relation between the two spans in the taxonomic literature of interest is coded in titles, sub-titles, variation in font type, font size, and spatial layout of the paragraphs. This introduces errors for the sentence splitting procedure in CoreNLP. The feature that we used to capture the relation (dependency grammar) is inherently limited since it is designed to work on relatively coherent, self-contained and complete sentences. Standard natural language processing tools are flexible and relatively easy to adopt, but for this particular problem it could be advantageous to use other, non-linguistic features to facilitate the information extraction, as has been suggested in the knowledge base creation literature (e.g., Schlichtkrull et al., 2018) but is outside the scope of our study.

We treat taxonomic ambiguities within TMO data in the same manner as OBIS and GBIF records (see section “DB data retrieval”).

From TMO location names to spatial data

Location names (e.g., New Zealand, Tasmania) are submitted to the Google geocoding service (https://developers.google.com/maps/documentation/geocoding/) to acquire a bounding box with four latitude-longitude coordinates and a centroid, based on Google’s defaults (Fig. S1). We remove species occurrences in locations represented by bounding boxes that are larger than about 2% of the Earth’s surface using area calculations assuming a spherical globe. The location names and their bounding boxes vary extensively in terms of their spatial resolution. The bounding boxes range from the scale of a few hundred meters across (for example “Fort Pierce Inlet”, in Florida, see Fig. S1), to major ocean basins like the “Mediterranean” or “Japan”. See Fig. S2 for how the bounding box sizes are distributed, and Fig. S3 for how the range-through diversity is affected when alternative cutoffs for the location size (lower or higher than ≈ 2%) are used.

Estimating latitudinal species richness

We initially evaluate species richness in thirty-six 5° latitudinal bands using two standard richness estimators that perform relatively well under a suite of conditions (Walther & Moore, 2005): Chao2 and Jackknife using the function specpool in the R package vegan (Oksanen et al., 2015). We treat these latitudinal bands as independent. We then repeat the procedure using thirty-six equal area bands, since areas represented within equal angle bands decrease poleward. To apply these estimators, we divide each (equal angle or area) latitudinal band into 5° longitudinal sampling units. We use the bias-corrected form of Chao2=Sobs+Q12(N−1)/(2NQ2), and incidence-based Jackknife = Sobs + Q1(N − 1)/N. Here, Sobs is the number of observed species in each band, N is the number of (longitudinal) sampling units, Q1 is the number of species observed in only one sampling unit, and Q2 is the number observed in two sampling units. The quantities Q1 and Q2 are to an extent sensitive to our choice of dividing the latitudinal bands into 5° units. The larger the bands, the higher chance that there will be more singletons, since the discrete state space is merged or reduced (see Fig. S7). Note that we measure incidence as whether a species is either observed or not observed, for each geographical unit. Hence, any duplicate records in OBIS and GBIF do not inflate richness. Because terrestrial regions are not suitable habitats for marine cheilostomes, we mapped all landlocked longitudinal sampling bins (Fig. S4) based on a 1:10 m map of global coastlines (Patterson, 2019). We removed the landlocked bins prior to richness estimation. For DB data where spatial coordinates are points, it is trivial to assign data to sampling units. For TMO, we assume that a species occurs in all of the sampling units (latitudinal bands, and longitudinal bins) that intersect the bounding box associated with the location. TMO bounding boxes vary in size, but most are smaller in area than our sampling units (Fig. S2).

In addition to Chao2 and Jackknife estimators, we also determined range-through species richness. Here, we assume that a species spans its southernmost and northernmost occurrence record, regardless of whether it is observed in any intermediate latitudinal band. We did not split the incidence data by hemispheres. If one species was present in the southern and northern hemispheres, we would consider it present in all latitudinal bands inbetween. We acknowledge that all richness estimators, including the ones we chose, have different limitations (Gwinn et al., 2016). Confidence that the inferred patterns are real is improved by the extent that different estimators making different assumptions yield consistent results.

Results

Capturing species richness: comparing DB and TMO

Applying the text-mining procedure to our corpora, we retrieved 1,408 species in 343 genera, and 1,400 unique location names among 7,204 TMO records. Only 23% of the species in the DB data that we retained were also in TMO. On the other hand, 68% of species in the TMO occurred in DB. 21% of the species richness is common to both (Fig. 1). In combination with DB data, we have species-location information from 4,910 species, almost tallying with the 4,921 described cheilostome species (Bock & Gordon, 2013).

Figure 1 Global range-through latitudinal species richness for cheilostome bryozoans.

The black line shows combined database (DB = OBIS and GBIF) and text-mined occurrence (TMO) richness, and orange and green curves show range-through richness for DB and TMO separately. The inset is a Venn diagram showing the global overlap in species between DB and TMO.

Our machine-classifier achieved an accuracy of 73.1%, F1 of 76.8%, recall of 78.9%, FPR of 34.3% and precision of 74.8% as estimated with our test set (Fig. S6b). These results are substantially better than a random classifier baseline, but not as good as the human annotator repeatability. Specifically, the FPR among annotators is about 15% (n = 200). A random classifier that is as unbalanced as our training data (60% positive labels) would yield 60% false positives, but a random classifier equaling our classifier’s recall of 78.9% would have the same false positive rate of 78.9% (see Appendix S1).

Latitudinal species richness patterns

Combined TMO and DB data in plots of range-through species richness show a bimodal pattern with species richness peaks in both hemispheres surrounding 40° and −40° (Fig. 1). Inferred species richness in both of these peaks is about double that in the tropics (Fig. 2). The two data sources contribute different latitudinal constituents, as suggested by the limited overlap in their species composition (Fig. 1 inset).

Figure 2 Global latitudinal species richness for cheilostome bryozoans, estimated using Chao2 and Jackknife.

The top panels (A & B) show richness for database (DB = OBIS and GBIF) and text-mined occurrences (TMO) data in 5° equal-angle latitudinal bands. The lower panels (C & D) show the equivalent in 5° equal-area latitudinal bands. Black lines show the observed richness, while blue and orange lines show the Chao2 and Jackknife estimates, respectively. The shaded areas are 95% confidence intervals. See Figs. S7 and S8 for alternative band and bin sizes.

Chao2 and Jackknife estimated species richness from DB shows two peaks between −20° and −45° that are more than double the next highest peak between 25° and 50° (Fig. 2A). In contrast, TMO estimated richness shows a highest peak between 30° and 45° (Fig. 2B). With minor exceptions in the Antarctic where spatial distortion is largest, equiangular and equi-areal bands yield nearly identical inferences (compare Figs. 2A, 2C). The latitudinal pattern appears smoother when using larger latitudinal band sizes (Fig. S7), but at the same time inflated, while retaining a qualitatively similar picture. Longitudinal sampling bins of varied sizes did not lead to notable variation for the Jackknife and Chao2 estimators (Fig. S8).

The northern hemisphere peak in richness (Fig. 1) and reflects TMO records from the Mediterranean and Japan, but also from the Atlantic Ocean (Figs. 3A, 3E), including the British Isles. Note that we did not include the Mediterranean as part of the Atlantic basin for Fig. 3. A portion of the TMO data are spatially imprecise, for example the location names “France”, “Spain” or “Morocco” may be associated with Mediterranean endemics, yet these records could contribute to the Atlantic richness counts in Fig. 3. The spatially precise DB data show a much lower peak in the Eastern Atlantic (Fig. 3E, orange line shifted slightly northward), reflecting data from the British Isles and northern Europe. Conversely, DB data mainly from Australia and New Zealand contribute disproportionately to the huge southern hemisphere peak. The richness captured by DB in Australia and New Zealand is not reflected by TMO species richness (Figs. 3B, 3D). The western Atlantic and eastern Pacific do not display such pronounced temperate zone peaks (Figs. 3C, 3F). Looking at individual ocean basins, TMO and DB are sometimes congruent and other times incongruent. For example, there is an absence of DB records in Japanese waters, and there are similarly few TMO and DB records in the Indian Ocean (Fig. 4).

Figure 3 Range-through latitudinal species richness for cheilostome bryozoans in the Atlantic and Pacific Oceans.

(A, C, D) Species richness in the Atlantic; (B, D, F) that in the Pacific. The panel rows represent the eastern, western or the entire ocean basins. Orange and green lines represent database (DB = OBIS and GBIF) and text-mined occurrences (TMO), respectively, and black lines are the joint data. Note that in this figure, the Atlantic borders Greenland and Iceland in the north, and the Antarctic in the south, but does not include the Gulf of Mexico, the Caribbean, the Baltic Sea or the Mediterranean. The Pacific borders the Bering Strait in the north, and includes the South China Sea, the Java Sea, north and east Australia, Tasmania as well as the Antarctic border.

Figure 4 Heatmaps for cheilostome bryozoan occurrence records per 5° latitude by 5° longitude bins.

The color axes are truncated for visualization purposes, to a maximum of 200, 200 and 2,000 in (A), (B), (C), respectively. (B) and (C) show the same sampling data, but in (C) the upper limit of the color axis is expanded by ten-fold. There are about 900 maximum records per bin in the Mediterranean for the text-mined occurrences (TMO), and about 66,000 maximum records in the British Isles for the Ocean Biodiversity Information System (OBIS) and Global Biodiversity Information Facility (GBIF) data combined. The globe is plotted using the Robinson projection. See Fig. S11 for the same figure plotted using the plate carrée projection.

Such varied regional species richness patterns are in part influenced by the geographic occurrence of samples. Figure 4 summarizes the relative distribution of species-location records for TMO and DB data as global heatmaps. For DB data, there are about one order of magnitude fewer records in tropical regions than for subtropical and temperate ones (Fig. S9a). While there are also fewer TMO records in tropical regions, the effect is not as pronounced (Fig. S9b). Northern and southern hemisphere species richness peaks in the two data sets (Figs. 3E, 3D) correspond with high regional densities of TMO and DB records, respectively (Figs. 3E, 3D).

Discussion

Causal hypotheses for a LDG and contrarian patterns are plentiful (Rivadeneira & Poore, 2020; García Molinos & Alabia, 2021). Such hypotheses can sometimes be tested in groups with rich and relatively unbiased spatial data from both extant and extinct taxa (Jablonski, Roy & Valentine, 2006; Krug, Jablonski & Valentine, 2007; Jablonski et al., 2013) or those with independent molecular phylogenetic evidence (Rabosky et al., 2018). We believe ours is the first study to quantify global cheilostome species biogeographic patterns. Using a combined TMO and DB perspective, and a bimodal latitudinal diversity gradient in cheilostome species richness is quite apparent. Yet, at present, we can merely speculate about what processes that may have led to their latitudinal pattern. Given the biases and heterogeneity of the data we explored which are striking when comparing our two data sources, we also need to consider (i) how this pattern coincides with previous observations, and (ii) methodological, sampling, and taxonomic concerns.

Two patterns in our analyses are similar to Schopf’s (1970) findings from then-scarce available data: higher species richness on the eastern margin of the Atlantic and the western margin of the Pacific compared to their opposite margins, and increasing richness with latitude away from the equator. Our combined data conforms with the first finding, but still does not capture the richness of the severely-understudied Philippine-Indonesian region and its many archipelagoes (Okada & Mawatari, 1953; Gordon, 1999; Tilbrook & De Grave, 2005). Changes to the second finding are more nuanced, and may partly reflect relatively lower equatorial sampling density (Menegotto & Rangel, 2018) apparent in both the datasets (Fig. S9). However, our observed peaks of species richness are at significantly higher latitudes than those reported for bryozoans in Chaudhary, Saeedi & Costello (2016).

Fossil and modern patterns of bryozoan skeletal abundance in cool-water carbonate sediments suggest that the lower tropical species richness is not merely a sampling artifact. Modern bryozoan-dominated carbonate platforms are far more common on cool-water temperate shelves than on tropical ones (Schlanger & Konishi, 1975; James & Clarke, 1997). Cenozoic tropical bryozoan faunas are both less abundantly preserved and less diverse than those from temperate latitudes, possibly reflecting biotic interactions, preservational biases, and cryptic existence in shallower-water habitats dominated by corals, calcareous algae, and other photobiont organisms (Winston, 1986; Taylor & Di Martino, 2014). A far-reaching study by Taylor & Allison (1998) showed that 94% of bryozoan-rich post-Paleozoic sedimentary deposits formed outside of the paleotropics, which may be especially significant if regional species richness and skeletal abundance are linked. About a third of all described bryozoan species occur south of −30°, and 87% of these are cheilostomes (Barnes & Griffiths, 2008).

We chose to discretize the data in latitudinal bands and longitudinal bins that are larger than those used previously (e.g., Rabosky et al., 2018). The choice of band- and bin sizes for species richness estimation is somewhat arbitrary. Differing choices suggest quantitatively dissimilar inferences, although the bimodality is still apparent in the cases we have explored (Figs. S7 and S8). A range-through latitudinal diversity approach (Fig. 3) assumes that any species that is not observed in a gap between two adjacent latitudinal bands should contribute to species richness in that gap, but this assumption is quite easily broken (Menegotto & Rangel, 2018). The bounding boxes used for TMO locations may also tend to bleed range margins as opposed to DB point location data. Richness estimates may be inflated via range-through estimates, particularly in the tropics, compared to estimating richness independently in each latitudinal band which yields lower estimates (Fig. 2). Regardless, both methods for estimating species richness give a picture of bimodality.

Global biogeographic studies such as ours are more prone to the issues of sampling and taxonomic concerns than local or regional ones, simply due to their scope. Large sampling gaps are apparent in both TMO and DB datasets. The development and application of richness estimation models that distinguish true absences from non-observations (Iknayan et al., 2014) may help improve inferences, but are likely insufficient to fully overcome acute sampling gaps. Overall, there are relatively few records in the Indian Ocean, most of the South Atlantic, and eastern margin of the Pacific. TMO records for the Arctic are sparse, as are OBIS records for the northwest Pacific. Aside from a few extreme outliers from DB British Isles locations, species richness and number of records per 5° latitudinal band have a strong positive relationship (Fig. S10). Independent taxonomic surveys of underrepresented regions in one or both datasets corroborate the existence of significant gaps (López Gappa, 2000; Barnes & Griffiths, 2008; Liu, 2008; Vieira, Migotto & Winston, 2008; Hirose, 2017; Boonzaaier-Davids, Florence & Gibbons, 2020; Denisenko, 2020; Sanjay et al., 2020). The DB records may partly reflect recent histories of active bryozoan research programs in the Antarctic (Barnes & Griffiths, 2008) and Australia and New Zealand (Wood et al., 2013) as well as contributions to OBIS and GBIF that differ substantially among research institutions. On the other hand, TMO extracted extensive species-location information from the Mediterranean (27° to 50°) that are severely wanting in OBIS, demonstrating that combining disparate data sources can help bridge gaps in global biogeographic studies.

Taxonomic errors inevitably exist in large databases. Taxonomy is continuously subject to revisions (Bock & Gordon, 2013), not all of which are accounted for in our datasets. Many species await description; Gordon et al. (2019) suggest that there are over 6,400 known cheilostome species without commenting on nomenclatural status, suggesting that there are up to 600 known species that need naming. Yet, a recent study, based on bryozoans, comparing datasets with taxonomic synonyms and without, found that synonymization does not contribute to qualitative changes in broad scale inferences (Lidgard et al., 2021). Our machine-classifier is currently unable to extract location information for 21% of the species that were detected in our corpus of published works (Fig. S5). Our conversion of taxonomic ambiguities into certainties likely deflated species richness estimates, while mistaken inclusion of fossil species names may have inflated richness estimates. We have assumed these do not necessarily introduce spatial bias. Additionally, many bryozoan species determined by traditional morphological methods may actually consist of unrecognized species complexes (Lidgard & Buckley, 1994; Fehlauer-Ale et al., 2014), although cheilostome bryozoan species are perhaps unusually delimitable using morphological information preserved in the skeleton (Jackson & Cheetham, 1990).

While the portion of TMO data that is derived from the taxonomic literature may be less plagued by taxonomic misidentifications, the same cannot be easily argued for faunal lists or ecological surveys, much of which DB data is based on. However, in our experience, broad inferences based on synoptic, large-scaled databases tend to change significantly with different models, more so than data updates (Sepkoski, 1993; Liow, Reitan & Harnik, 2015; Lidgard et al., 2021).

In terms of our text-mining task, we found that generating and classifying species-location candidates here is more challenging than classifying species-age candidates (Kopperud, Lidgard & Liow, 2019). An F1 result of about 77.5% is not uncommon for relation extraction studies (Kim, Kim & Lee, 2019; Henry et al., 2020), especially for datasets with low label assignment repeatability. While the accuracy of the machine-classifier is less sensitive than human evaluation, its FPR is substantially lower than a null model. Yet, it remains a fact that the number of false positives are substantially higher than we would have liked. We can not exclude the possibility that the false positives had a confounding effect on the diversity estimates in the northern hemisphere. It essential to recognize that the classifier merely provides a probabilistic measure of whether the sentence provides evidence that a species is present at a geographic location. In the event of a false positive, it is still possible that the species is actually present in that particular location. On the other hand, there is a wealth of species mentions for which we were not able retrieve any species-location candidates (Fig. S5). It is possible to extend our approach by considering cross-sentence candidates (Gupta et al., 2019), although these methods are usually less accurate.

One caveat of using natural language for information extraction, is that the available tools are by far most developed for English. There is rudimentary support for widely used languages such as Mandarin, Spanish and French, however it would be more difficult to perform a relation extraction task for these. Using our approach, we would need to re-train the classifiers specifically for each language. This entails a duplication of work, which considering how most modern articles in bryozoology are in English, is likely not worth the effort. Language-agnostic approaches to relation extraction exist (Heist & Paulheim, 2017), however these methods can be more challenging to apply for a specific domain problem like ours. Nevertheless, the fact remains that we only looked at English literature, and this may contribute to spatial or taxonomic biases in our understanding of the latitudinal gradient. Alternatively, we could go beyond standard NLP tools, which are relatively flexible and easy to adopt, and use non-linguistic features such as tables and spatial layout common in primary diversity publications (e.g., Rosso & Sanfilippo, 2000; Gordon, 2016) for information extraction, as has been suggested in the knowledge base creation literature (Schlichtkrull et al., 2018). However, such methods for information extraction that combine linguistic and non-linguistic features are still at an early stage of development.

The main advantage of automatic information retrieval over collaborative data-entry is that of reduced time and resource investment. The information retrieval procedure is largely independent of the size of the literature, or the taxonomic scope, say for cheilostomes vs. all metazoans. Public biodiversity inventories such as GBIF and OBIS require large consortia and networks of research factions to contribute their data. Conversely, there is a wealth of biodiversity knowledge available in the published literature, and it is feasible for one person or a small team to extract substantial amounts of data quickly using automated information retrieval. We have used some supervised classification methods, which require us to generate training data. However as NLP is adopted in the biodiversity literature, it will become easier to use distantly supervised relation extraction (Hirschberg & Manning, 2015).

Biodiversity inventories such as OBIS and GBIF are vital for supplying data for inferences of global biogeographic patterns. While we strongly support the continued development of these databases, we demonstrated that our automated information retrieval approach can enhance such inventories when answering global-scale questions, especially for under-studied taxa. To understand how the spatial diversity of cheilostomes has come to be will require continued and concerted efforts in taxonomic investigations (Bock & Gordon, 2013), compilation of more spatial data especially in areas currently devoid of deposited information (Klein et al., 2019), tool-development in automated data retrieval (Kopperud, Lidgard & Liow, 2019), and continued research in molecular phylogenetics (Orr et al., 2021).

Supporting information

Appendix S1: Extended methods and supplementary figures.

Appendix S2: Bibliographic references for TMO data.

The code and data required to reproduce the analyses and figures is available at www.zenodo.org/record/6770200 (DOI 10.5281/zenodo.6475658).

Supplemental Information

Supplemental Information 1 Supplementary text and figures.

Click here for additional data file.

Supplemental Information 2 References used in text-mining.

Click here for additional data file.

We thank Mali H. Ramfjell for compiling part of our training dataset, the GeoDeepDive group, especially Ian Ross and Shanan Peters, for providing access to articles, and OBIS and GBIF and their contributors for their georeferenced taxonomic data. We thank Phil Bock for maintaining bryozoa.net, Dennis Gordon for an updated version of the Working List of Genera and Subgenera for the Treatise on Invertebrate Paleontology, and both for their contributions to WoRMS.

Additional Information and Declarations

Competing Interests

Author Contributions

Data Availability

The authors declare that they have no competing interests.

Bjørn Tore Kopperud conceived and designed the experiments, performed the experiments, analyzed the data, prepared figures and/or tables, authored or reviewed drafts of the article, programming for NLP, and approved the final draft.

Scott Lidgard conceived and designed the experiments, performed the experiments, authored or reviewed drafts of the article, and approved the final draft.

Lee Hsiang Liow conceived and designed the experiments, performed the experiments, authored or reviewed drafts of the article, and approved the final draft.

The following information was supplied regarding data availability:

The raw data is available at Zenodo: Kopperud, Bjørn Tore, Lidgard, Scott, & Liow, Lee Hsiang. (2022). Enhancing georeferenced biodiversity inventories: automated information extraction from literature records reveal the gaps [Data set]. In PeerJ. Zenodo. https://doi.org/10.5281/zenodo.6770200.

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
