# Peer review of "Enhancing georeferenced biodiversity inventories: automated information extraction from literature records reveal the gaps"

_PeerJ, doi:10.7717/peerj.13921_

## Round 0.1 · original submission · Major Revisions

I apologize for the slow return of reviews. Reviewer 1 makes a number of useful points and raises the concern of the high-rate of false positives. The reviewer also suggests looking further into the actual causes of the differences between the TMO and DB latitudinal richness distributions. Reviewer 2 has a more limited review but does make some useful recommendations. One unfortunate event was that this reviewer was not able to decompress the supplementary data. I would request that you deposit the data using a widely used compression system (like .zip) that is easily navigated by potential users and reviewers. Your paper may or may not be sent out for a second round of review, depending on the quality of your responses to the reviewers' comments.

·

Basic reporting

See Additional Comments below for overall review summary – here specific criteria addressed in bullet form.

Major Comments:
-Paper is well-written in easily-readable English, reporting is overall of high quality.
-In the methods section, significantly more detail is needed about the procedure of TMO data retrieval. Right now, it is essentially presented as a black box with only a heuristic example given for a single sentence. There is clearly a lot more methodological information to include, given the length of the associated methods descriptions in the supplementary material. Some of the information should be pulled from the supplement into the main text, from sections ii-v of “Automated retrieval of location-species data”.

Minor Comments:
-The introduction section titled “Automated information retrieval” could be expanded significantly to include more background on natural language processing, especially given that the audience of the paper is likely to be primarily biodiversity scientists.
-The order of the sections “Cheilostomes” and “Automated information retrieval” could be switched for overall better flow.
-The color palette of Figure 4 is a bit confusing. I understand the use of four different colors to make the levels easier to tell apart, but the sequence of purple, green, orange, blue as being low to high is not very intuitive or continuous. Additionally, the green and orange levels could potentially cause problems for color-blind people? Perhaps a better sequence would be gray, purple, blue, green, yellow?
-Would be nice in the discussion to discuss the potential of NLP for overcoming language barriers in accessibility of scientific data. I.e., sampling bias due to looking at publications only in English.

Experimental design

Major Comments:
-Overall design and structure of the study are strong.
-I do think that there is a crucially untapped opportunity to investigate the actual causes of the differences between the TMO and DB latitudinal richness distributions. The methods section is full of descriptions of these differences across methods, but the closest thing to an explanation the readers ever get is the attribution of some of the patterns to certain robust national datasets. An extended exploration of the following questions in the discussion section would greatly strengthen the paper:
1) Why exactly do you expect the TMO and DB distributions to differ (in the introduction it is stated that “Taxon occurrence data from DB and TMO are not expected to be the same” with no justification), and what are the different processes (in data gathering, processing, publishing, etc.) that might cause them to differ?
2) What are the different forms of sampling bias (if they differ it surely must be because of sampling bias since there is only one true diversity distribution)?
3) Can combining the two data sources overcome certain types of sampling bias? Are the two data sources complementary in their information?

Minor Comments:
-None

Validity of the findings

Major Comments:
-I am concerned about the 34.3% false positive rate for candidate species location relations. This is a large number of false positives (even if it is better than a random classifier, which is not a particularly high bar), and surely introduces a lot of spurious occurrence locations that might skew the detected latitudinal diversity signal. If I understand correctly, with a precision of 74.8% you are approximately looking at a quarter of the confirmed species-location relations being false, and I am worried that the observed TMO latitudinal richness gradient is reflecting the overall distribution of location names in the database rather than only true observation locations. Could you compare your TMO latitudinal richness gradient with some sort of null? For example, with random classification, what does the latitudinal gradient look like? Similarly, what is the latitudinal distribution of location names in the text in general? Does not have to be done this way, just some ideas.
-The confidence intervals around the Chao2 and jacknife estimators for the TMO latitudinal richness gradient are very tight. Is there any reason to expect that the TMO species occurrence data should yield more confidence in predictions? If so, it should be explained in the text. Would it be possible that it is the result of a form of pseudoreplication by which the same single occurrence may be referred to by location multiple times in the same text, or even across texts (citations of previous studies)? In which case the number of singleton observations for the chao2 indicator would be artificially suppressed, and the uncertainty artificially low. How the algorithm treats these situations are also the sort of details that should be expanded in the methods.
-When you calculate range-through species richness, are you applying the method even for species that span both sides of the equator? It seems a sensible thing to do when the middle location is environmentally between the extremes, but wouldn’t work for species that for example are observed at 40N and 40S and are inferred to inhabit equatorial ocean. If it is only done within hemisphere, it should be documented.

Minor Comments:
-It would be nice to get some information characterizing the overall georeferencing noise. You exclude georeferences by land area, but you still might get some edge cases where the latitude range is incredibly broad – I’m thinking in particular countries such as Chile or Japan. The reader should have a rough idea of how much uncertainty there is associated with georeferencing.

Additional comments

In this study, Kopperud, Lidgard, and Liow explore the use of natural language processing for the automated generation and georeferencing of species occurrence records from academic literature databases. The authors focus on cheilostome bryozoans, and combine text-mined occurrences with more traditional species occurrence records to uncover data source-dependent patterns of latitudinal bryozoan species richness. The study is exciting and well-conceived; text-mined occurrences could potentially become an invaluable source of information for data-limited taxa, and the study of macroecological patterns such as latitudinal species richness represents a logical first step in evaluating the utility, but also limitations of these automatically-generated occurrences. However, the current manuscript contains a number of methodological limitations related to text-mined occurrences that must at least be addressed, if not solved, before the article is suitable for publication. These are listed above, but the foremost of these is the relatively high false positive rate (34.3%) of accepted candidate species-location relations. With an approximately 3:1 ratio of true observations to false positives (74.8% precision), the TMO latitudinal richness gradient presented seems like it could be skewed by the text-mining methodology. Overall though, the objective of the study is quite exciting, and I believe if the authors address the major considerations above it would be a very valuable publication. I look forward to reading the authors’ responses.

·

Basic reporting

Language is clear and professional
Intro and background show context but lines 84 – 88 seem like results rather than introduction.
As far as I can tell structure conforms to PeerJ standards.
Figures are well done. For Figure 4 it is not clear to me why locations in the heatmap disappear between b) and c) when the number of max records is adjusted from 200 to 2000.

Experimental design

Most of the experimental design I am not knowledgeable enough to comment on. I hope PeerJ has found someone with a background in NPL to also review the paper. DB Data Retrieval is appropriate and their use of the data is appropriate. Disregarding the taxonomic ambiguities lowers the taxonomic range for the analysis but they address this in the text. I cannot comment on the TMO Data Retrieval. I believe the way they have estimated latitudinal species richness is appropriate although I did have a question about how bounding boxes were used in the designation of a single TMO occurrence to multiple latitudinal bands. I think the paper would be improved with a longer explanation of how these broad spatial “observations” may have affected the analysis and results. I cannot comment on the use of Chao2 or Jackknife. I hope the editors have found someone to review these portions as they are outside the scope of my expertise.

Validity of the findings

The findings look valid to me although as mentioned before in the experimental design I would like a more detailed explanation of how the bounding box TMO “observations” could have affected the results.

Additional comments

1. Data from OBIS and GBIF are not appropriately cited. Authors need to follow the data citation guidelines for these systems. In particular I recommend using the derived dataset citation capability supported by GBIF for all the GBIF records so that data providers can be appropriately credited for the use of these observations in the analysis. https://www.gbif.org/derived-dataset/about
2. OBIS has undergone a name change and is now the Ocean Biodiversity Information System. This needs to be corrected throughout the text. Also the web address for OBIS is www.obis.org.

---

## Round 0.2 · accepted · Accept

Please ensure that the very minor comments of both reviewers are addressed at the proof stage. Congratulations on an excellent contribution to PeerJ and biodiversity science!

·

Basic reporting

The additions to the introduction and methods from the authors are helpful and appreciated! I will note that for figure 4, I was not requesting a shift to a continuous color gradient, but just a reordering of the discrete color levels to a more intuitive sequence. But it is not a big issue, fine to keep as is.

Experimental design

Thanks to the authors for addressing the concerns - I'm not sure if I agree that sampling bias is an unrelated/tangential topic to the main message of the paper though! Perhaps 'bias' is not the right word here, but I do think there is much to be explored about the differences in the fundamental data collection processes that influence how information ends up in databases or publications, and how those processes might differ and result in the very different latitudinal diversity gradients you are getting between DB and TMO. Presumably, different mechanisms of data collection, aggregation, and publication are at the root of your observed differences in DB and TMO? Those differences in processes are really interesting to me in their own right, are what I meant when I said sampling bias between DB and TMO - sorry if it wasn't clear before! But although I'd ideally like to see this facet expanded it is not critical, and happy with it as is.

Validity of the findings

Thanks again to the authors for addressing the concerns. It is too bad that not much can be done about the false positive rate, but thanks for improving the documentation on that front. Thank you for elucidating the difficulty of creating a null for the TMO distribution - it makes sense to me and seems tricky indeed. All fine left as is.

Additional comments

The authors did well to address my previous concerns. I have added some comments above, but the manuscript is overall suitable for publication. Well done to the authors!

·

Basic reporting

The edits made in the text make it clearer the differences in Figure 4. Thank you.

Experimental design

Edits to the text make it clearer how the bounding boxes are used in the analysis.

Validity of the findings

No further comments. Issues were addressed above.

Additional comments

1. I was able to download the zip file from Zenodo, unzip it, and access all the files. Thank you for making this change.
2. Thank you for citing the data.
3. The OBIS name change is not addressed in all sections of the manuscript such as the abstract and line 41. I would appreciate if the copy editors for the journal will ensure this is consistent throughout the manuscript. It should be the Ocean Biodiversity Information System.